# Selective Control by *Pistacia vera* L. and Its Carotenoid Zeaxanthin on SARS-CoV-2 Virus

**DOI:** 10.3390/ijms26041667

**Published:** 2025-02-15

**Authors:** Rosamaria Pennisi, Davide Gentile, Paola Trischitta, Davide Barreca, Antonio Rescifina, Giuseppina Mandalari, Maria Teresa Sciortino

**Affiliations:** 1Department of Chemical, Biological, Pharmaceutical and Environmental Sciences, University of Messina, Viale Ferdinando Stagno d’Alcontres 31, 98166 Messina, Italy; paola.trischitta@studenti.unime.it (P.T.); davide.barreca@unime.it (D.B.); gmandalari@unime.it (G.M.); 2Department of Chemistry, Materials and Chemical Engineering “G. Natta”, Politecnico di Milano, Via Mancinelli 7, 20131 Milano, Italy; davide.gentile@polimi.it; 3Department of Chemistry, Biology and Biotechnology, University of Perugia, Via Elce di sotto 8, 06123 Perugia, Italy; 4Department of Drug and Health Sciences, University of Catania, V. le A. Doria, 95125 Catania, Italy; antonio.rescifina@unict.it

**Keywords:** SARS-CoV-2, natural antivirals, pseudovirus, pistachio extracts, TMPRSS2, 3CL^pro^

## Abstract

Since the onset of the COVID-19 (COronaVIrus Disease 19) pandemic, SARS-CoV-2 has exhibited a high transmission rate, further enhanced by new variants able to better adapt to humans. Addressing this issue has been challenging due to viral resistance and side effects associated with antiviral drugs and vaccines. As a result, there has been a growing interest in plant-derived compounds with antiviral properties. Our study revealed that pistachio extracts significantly inhibited SARS-CoV-2 viral entry. Employing pseudotyped particles bearing the S protein of SARS-CoV-2, we demonstrated that treatment with pistachio extracts inhibited binding of alpha (α) and omicron (ο) SARS-CoV-2 variants. Furthermore, our study revealed that the pistachio carotenoid zeaxanthin exhibited a different inhibitory activity against two SARS-CoV-2 variants. In silico analyses demonstrated a strong interaction between zeaxanthin and the receptor-binding domain (RBD) domain of the omicron spike (S) protein, thus reducing pseudovirus entry. However, zeaxanthin’s weaker interaction with the alpha variant’s RBD was insufficient to inhibit entry. Additionally, zeaxanthin suppressed the expression of the host protease TMPRSS2 at the protein level, thereby limiting the internalization of the alpha variant, which relies on TMPRSS2 for cellular entry.

## 1. Introduction

The continuous emergence of respiratory viruses with pandemic potential underscores the critical need for alternative antiviral treatments. Over the past 20 years, zoonotic transmission has led to the emergence of highly pathogenic strains of coronaviruses (CoV) in human populations, including SARS-CoV, MERS-CoV and, recently SARS-COV-2, the etiological agent of COVID-19. According to the International Commission on Taxonomy of Viruses (ICTV, 2021), SARS-CoV-2 is classified within the subgenus Sarbecoronavirus, genus Betacoronavirus, sub-family Orthocoronavirinae, family Coronaviridae, order Nidovirales, phylum Pisuviricota, kingdom Orthornavirae, domain Riboviria.

Coronaviruses, particularly alpha and beta coronaviruses, typically infect mammalian species and cause mild seasonal infections affecting the upper respiratory system [1,2]. SARS-CoV-2, like other human coronaviruses, contains four structural proteins (S, E, M, N) and sixteen non-structural proteins (NSP1-16), essential for RNA replication, transcription, and immune evasion [2,3,4]. The spike (S) protein, composed of S1 and S2 subunits, is crucial for the virus’s ability to enter host cells. The S1 subunit binds to the cellular receptor angiotensin-converting enzyme 2 (ACE2), while the S2 subunit facilitates membrane fusion [5,6,7,8].

This dual function of the spike protein underscores its importance in the viral life cycle and identifies it as a significant therapeutic target. By inhibiting the interaction between the S1 subunit and ACE2 or blocking the fusion process mediated by the S2 subunit, potential treatments can effectively prevent the virus from infecting host cells, thereby halting the progression of the infection. ACE2, identified as a functional host receptor for SARS-CoV-2, is abundant in various cell types, particularly in type 2 alveolar cells and the nasal epithelium [9,10]. The altered expression of ACE2 is associated with the severity of COVID-19, especially in individuals with cardiovascular diseases and metabolic syndrome [11].

SARS-CoV-2 entry into host cells is facilitated by the cleavage of the spike protein, either by TMPRSS2 (Transmembrane Serine Protease 2) on the cell membrane or by cathepsin L within endosomes [8]. Variants of concern (VOC) have emerged, characterized by mutations that affect transmissibility, pathogenicity, and immune escape. For example, the Alpha variant (B.1.1.7) has mutations that stabilize the spike-ACE2 complex, increasing infectivity and mortality [12]. The Omicron variant, with over 30 mutations in the spike protein, shows significant immune escape and higher transmissibility [13,14,15] and prefers a TMPRSS2-independent mechanism to enter host cells [16] efficiently.

Considering the emergence of SARS-CoV-2 variants and the virus’s persistence, which challenge the current vaccines’ effectiveness, the search for additional antiviral agents remains critical. However, traditional drug development approaches can take decades and cost billions of dollars, which significantly complicates the process of identifying new antivirals against current and emerging mutated strains [14].

In this context, natural products and dietary compounds have gained attention for their potential antiviral properties, offering a complementary approach to traditional pharmaceuticals [17,18]. Pistachio extracts and zeaxanthin, a carotenoid found in various fruits and vegetables, have been previously noted for their antiviral, antioxidant, anti-inflammatory, and immunomodulatory potential [19].

These properties could play a role in inhibiting viral replication and enhancing immune responses. However, their inhibitory effect against SARS-CoV-2 has not been thoroughly investigated. The present study aimed to explore the antiviral potential of pistachio extracts and its carotenoid zeaxanthin against SARS-CoV-2, unveiling the mechanism responsible for the observed effect. Therefore, in vitro experiments using SARS-CoV-2 pseudotypes, which mimic the virus-host cell interactions, and the A549 cell lines, commonly used for the study of respiratory infections, were employed to evaluate the efficacy of pistachio extracts and zeaxanthin in preventing viral entry.

In silico studies were employed to investigate the molecular interactions between natural compounds and the spike proteins of the alpha and omicron variants of SARS-CoV-2. This computational approach provides valuable insights into the inhibitory mechanisms of these compounds and their therapeutic potential against SARS-CoV-2 and emerging variants.

## 2. Results

### 2.1. Inhibition of SARS-CoV-2 Variants Binding by Pistachio Extracts

In order to investigate the role of pistachio extracts as potential entry inhibitors of SARS-CoV-2, we employed a safe and convenient assay by generating a luciferase (Luc)-expressing pseudovirus containing the wild-type or mutant S protein of SARS-CoV-2 in the envelope-defective lentiviral backbone (Appendix A). The experimental procedure aimed to generate coronavirus (CoV-2) spike (S) fusion protein pseudotyped particles using a Lentivirus core and luciferase reporter in a transient transfection of HEK-293T cells. The construction of SARS-CoV-2 pseudoviruses is detailed in Material and Methods [20]. The efficiency of yielded-pseudotyped particles was measured by gene reporter expression (Appendix A). Then, we evaluated the compounds’ cytotoxicity on A549 cells to determine the CC_50_ value (Appendix A). This value is essential in establishing a safe and effective dosage for the antiviral treatment without causing undue cell harm. Thus, α- and ο-SARS-CoV-2 pseudotypes were pretreated with pistachio extracts identified as NRRE (Natural raw polyphenols-rich extract) and RURE (Roasted unsalted polyphenols-rich extract) [21], detailed in Material and Methods, for 1 h at 4 °C and used to transduce the A549 cells.

The inhibitory binding activity of both NRRE and RURE on the two SARS-CoV-2 pseudotypes was assessed by measuring the luciferase activity, and the results were reported as relative luciferase units (RLU) and compared to untreated cells. The procedure is graphically reported in Figure 1 Panel a. The results showed that both extracts reduced the entry of α (Figure 1, panels b and c) and ο-SARS-CoV-2 pseudoviruses (Figure 1, panels d and e).

### 2.2. Inhibition of SARS-CoV-2 Variants by Zeaxanthin

Based on previous data obtained in our laboratory demonstrating that zeaxanthin, a component found in both NRRE and RURE, exhibited antiviral activity against HSV-1 by blocking its entry [19], we investigated whether zeaxanthin could exert a similar inhibitory effect on SARS-CoV-2 by preventing its binding to host cells. To this end, first, we tested zeaxanthin cytotoxicity on A549 cells to determine the CC_50_ value (Appendix A). Then, we pretreated pseudotype variants with zeaxanthin for 1 h at 4 °C, a temperature that typically prevents fusion/entry but is not binding. After pretreatment, we transduced A549 cells with both pseudotype variants for 2 h. This procedure is graphically illustrated in Figure 2a. The data obtained revealed that zeaxanthin significantly inhibited the binding of the ο-SARS-CoV-2 but not the α-variant (Figure 2, panel b vs. c). This different inhibition highlights the specificity of zeaxanthin’s interaction with SARS-CoV-2 variants and may be attributed to structural differences in the receptor-binding domain (RBD) of these variants. In silico analyses were carried out to elucidate the mechanism underlying this different activity (Figure 3).

### 2.3. Docking and Molecular Dynamics Simulations

The crystallographic structures of the RBDs in complex with ACE2 were used to investigate the interactions between the spike proteins of the alpha and omicron variants and zeaxanthin. A detailed analysis of the two-dimensional interaction diagram (Figure 3) revealed that zeaxanthin engages in hydrophobic interactions with the RBD alpha variant (Appendix A). Specifically, the isoprene units of zeaxanthin interact with Ala386, Ala387, and Pro389, while the aliphatic ring interacts with His34, Pro321, and Met383 (Figure 3a,b). Additionally, two hydrogen bonds were identified between the hydroxyl groups at the terminus of zeaxanthin’s aliphatic chain and the residues Lys26 and Met323.

Molecular dynamics (MD) simulations were conducted over 200 ns to elucidate further zeaxanthin’s different behavior with the alpha and omicron variants. For the zeaxanthin/α-RBD complex, the root-mean-square deviation (RMSD) of the Cα atoms exhibited an initial variation of up to 2.6 Å within the first 50 ns, after which it stabilized for the remainder of the (Appendix A).

In contrast, the RMSD of the ligand itself displayed significant fluctuations, reaching up to 14 Å within the first 15 ns and stabilizing between 10 and 12 Å for the rest of the simulation. Throughout the MD simulation, the interactions between zeaxanthin and the α-RBD were predominantly hydrophobic, while hydrogen bonds were notably unstable, frequently disappearing over extended periods (Appendix A). This behavior may explain the in vitro observations, suggesting that zeaxanthin forms transient and inherently unstable interactions with the α-RBD. In the case of the ο-RBD variant, the best docking pose of zeaxanthin revealed that the ligand was deeply buried within a subunit adjacent to the ACE2 interaction domain (Appendix A). The interaction diagram (Figure 3c,d) highlighted extensive hydrophobic interactions with residues Pro321, Met383, Phe356, Ala386, Ala387, Pro389, Pro321, Leu29, Val93, and stable hydrogen bonds with Met323, Aspe30, and Lys26 at distances of 1.86, 2.98, and 2.34 Å, respectively.

Analysis of the MD trajectories for the zeaxanthin/o-RBD complex showed a distinct behavior compared to the α-variant. The ligand RMSD reached equilibrium at approximately 4.5 Å after an initial fluctuation within the first 50 ns (Appendix A). Notably, the complex exhibited greater stability, as evidenced by more hydrogen bonds throughout the simulation (Appendix A). This stability contrasts sharply with the behavior observed in the α-RBD complex.

After 200 ns of simulation, zeaxanthin within the α-RBD exhibited significant displacement, positioning itself far from the ACE2 interaction region (Figure 4a). Analysis of the final frame of the MD simulation revealed hydrophobic interactions with Leu560, Arg559, Ala387, Phe356, and Met383, along with a single hydrogen bond with Glu329 (Appendix A). In contrast, the zeaxanthin/ο-RBD complex maintained two stable hydrogen bonds with Asp30 and Asn322, alongside hydrophobic interactions involving the isoprene chains and residues His34, Pro389, Ala386, Ala387, Phe356, Met383, and Pro321 (Appendix A).

### 2.4. Inhibition of SARS-CoV-2 Variants by Zeaxanthin upon Cell Treatment

To investigate whether zeaxanthin exhibited activity against TMPRSS2, which is selectively involved in the entry of the alpha variant but not the omicron variant, we conducted an experiment where cell monolayers were pretreated with zeaxanthin before the transduction of pseudoviruses. Specifically, A549 cells were pretreated with zeaxanthin for 2 h at 37 °C, as illustrated in Figure 5a. This was followed by the transduction with α and ο-SARS-CoV-2 pseudotype variants separately, allowing us to observe the effect of zeaxanthin on cells affecting the virus entry. The results showed that zeaxanthin inhibited the alpha entry but not the omicron variant (Figure 5b vs. Figure 5c), suggesting that it could affect the TMPRSS2-mediated internalization.

This has significant implications for potential antiviral therapeutics, offering a ray of hope in the fight against the SARS-CoV-2 new variant different from omicron or against other viruses that use cellular proteins such as TMPRSS for entry [15].

### 2.5. Inhibition of TMPRSS2 by Zeaxanthin upon Cells Treatment

To acquire data on zeaxanthin’s mechanistic activity, we tested the zeaxanthin inhibitory activity on cellular proteases TMPRSS2, which is involved in viral entry. First, we treated the A549 cells with or without zeaxanthin for two hours, and then we transduced treated cells with α and ο-SARS-CoV-2 pseudotype variants separately. We analyzed the levels of TMPRSS2 protein accumulation using western blot analysis. The findings show that treatment with zeaxanthin reduced the accumulation of TMPRSS2 protein alone or upon the transduction of the α- SARS-CoV-2 pseudotype variant (Figure 6). The degree of the TMPRSS2 accumulation was less evident following the treatment and transduction with the ο-SARS-CoV-2 pseudotype. This suggests that zeaxanthin treatment can reduce TMPRSS2 accumulation in A549 cells. Furthermore, transduction with the α-SARS-CoV-2 pseudotype appears to have a more significant impact on TMPRSS2 protein levels. The above data can justify the failure to use TMPRSS2 by the omicron SARS-CoV-2 variant for entry in comparison with the alfa SARS-CoV-2 variant [21]. These data are supported by our previous experiments confirming efficient cathepsin-mediated entry by omicron SARS-CoV-2 pseudotype variant in a monkey kidney cell line [20].

### 2.6. In Vitro Screening for 3CL^pro^ and PL^pro^ Inhibition by Zeaxanthin

Proteases have been identified in DNA or RNA viruses and enveloped and non-enveloped viruses. These enzymes can use different catalytic mechanisms involving Ser, Cys, or Asp residues to attack the cleavable peptide bond with a high degree of recognition of specific cleavage [15]. Given the role of zeaxanthin in the modulation of cellular protease TMPRSS2, we performed an in vitro screening for 3CL^pro^ and PL^pro^. 3CL^pro^ (also known as Main Protease or M^pro^) and PL^pro^ are chymotrypsin-like proteases encoded in the SARS-CoV-2 genome and play essential roles in the virus’s lifecycle. A luminescent protease assay was carried out to verify the anti-protease activity of zeaxanthin (Figure 5). Recombinant SARS-CoV-2 3CL^pro^ and PL^pro^ were separately combined with luminogenic 3CL^pro^ substrate and zeaxanthin for 60 min at 37 °C. The reaction was stopped by adding Luciferin Detection Reagent, and the luminescence was recorded on a plate-reading luminometer. The GC376 was used as a positive control. The results expressed as RLU and percentage of inhibition of enzymatic activity (Figure 5, panels a and c) showed that the enzymatic activity of 3CL^pro^ was inhibited following incubation with zeaxanthin. Similarly, zeaxanthin significantly reduced the activity of papain-like protease PL^pro^, as Figure 7b,d reports.

## 3. Discussion

The present study explored the potential of pistachio extracts (NRRE and RURE) and zeaxanthin as inhibitors of SARS-CoV-2 entry, utilizing a luciferase-expressing pseudovirus system. NRRE and RURE extracts demonstrated significant inhibition of entry for both α- and ο-SARS-CoV-2 pseudoviruses, as shown in Figure 1. The natural product zeaxanthin, previously shown to inhibit HSV-1 entry, also acted as a SARS-CoV-2 entry inhibitor, although the targets and mechanisms differ [19].

Molecular modeling studies demonstrated that zeaxanthin strongly interacts with the RBD of the omicron variant’s spike protein (Figure 3 and Figure 4). These findings correlate with the observed reduction in viral entry in in vitro studies (Figure 2), suggesting that zeaxanthin may play a role in inhibiting viral infection. The combination of molecular docking and MD simulations provides a robust computational framework for in silico analysis of drug-protein interactions, offering critical insights that complement and validate in vitro experimental results [22]. To evaluate the potential of zeaxanthin in modulating spike RBD-ACE2 interactions, we employed the Autodock4-GPU docking software (v. 4.2.6). Zeaxanthin was docked to the RBD of the SARS-CoV-2 spike protein of the alpha and omicron variants. While the binding energies were comparable (−6.90 kcal/mol for the alpha variant and −7.10 kcal/mol for the omicron variant), the 200 ns MD simulations revealed distinct behavioral differences between the two variants. The best docking poses of zeaxanthin with the RBD domain highlighted the different stability of the complexes. Specifically, the zeaxanthin/ο-RBD complex exhibited greater stability compared to the alpha variant. In the case of the omicron variant, zeaxanthin remained stably positioned within the RBD region throughout the 200 ns simulation, as evidenced by minimal RMSD fluctuations. The proximity of zeaxanthin to the RBD/ACE2 interaction interface suggests that the compound may destabilize the protein complex, thereby hindering viral entry. This hypothesis is further supported by the free energy of binding analyses conducted using MM-PBSA calculations over the 200 ns simulation. According to YASARA analysis, more negative free energy of binding values indicates stronger binding affinity. The results showed average free energy of binding of −6.90 kcal/mol for the α-RBD complex and −8.77 kcal/mol for the ο-RBD complex. Additionally, the docking-based method used to calculate binding energy during the MD simulation indicated a higher affinity for the ο-RBD complex (−4.86 kcal/mol) compared to the α-RBD complex (−4.34 kcal/mol). These findings and the MD simulation results underscore the more favourable interactions in the omicron complex (Appendix A).

The different inhibitory effects of zeaxanthin on SARS-CoV-2 variants likely stem from structural variations within the RBD of these variants. This hypothesis is supported by a previous study investigating the binding affinities of various carotenoids, including zeaxanthin, with SARS-CoV-2 spike protein mutants [22]. Although the omicron variant was not included in that study, the authors observed variant-dependent binding affinities, with zeaxanthin exhibiting stronger binding to the delta variant and weaker binding to the alpha variant. These results reinforce the notation that sequence differences in the spike protein’s RBD significantly influence the binding efficiency of zeaxanthin across SARS-CoV-2 variants.

It is well known that SARS-CoV-2 entry into respiratory cells is facilitated by the interaction between the viral spike protein and the ACE2 (angiotensin-converting enzyme 2) on the surface of lung cells, followed by membrane fusion that allows the viral nucleocapsid to enter the cell. TMPRSS2 (Transmembrane Serine Protease 2) plays a crucial role in the entry process of SARS-CoV-2 into human cells but less in the omicron variant. TMPRSS2 cleaves the S protein at specific sites, activating it. The activation of the spike protein by TMPRSS2 is essential for the virus to bind to the host cell receptor ACE2 (Angiotensin-Converting Enzyme 2) and subsequently fuse with the cell membrane.

Building on these data, we identified a carotenoid, zeaxanthin, present in small quantities in NRRE and RURE, as a potential key factor responsible for the observed antiviral activity following pistachio extract treatment. Our investigation into the effect of zeaxanthin on the cellular protease TMPRSS2 revealed that it inhibited the entry of the α-variant but not the ο-variant. This was supported by reduced TMPRSS2 levels observed by western blot analysis in the presence of zeaxanthin. Our results showed that zeaxanthin displays a potent inhibitor effect on TMPRSS2, specifically hampering the α-SARS-CoV-2 variant, which used it for entry, without affecting the o-SARS-CoV-2 variant (Figure 6).

The data seem useless, given that the omicron variant is currently the most widespread worldwide. Nevertheless, as mentioned in [15], all the prominent animal viruses, herpesviruses, HCV, alphaviruses, flaviviruses, HIV-1, and picornaviruses contain proteases that play crucial roles in their replication and thus, are essential targets for discovering potent antiviral drugs. Therefore, our data indicate for the first time that zeaxanthin can act as a natural inhibitor of TMPRSS2. This finding adds zeaxanthin to a growing list of natural compounds with potential anti-viral properties through the inhibition of key proteases involved in viral entry [23,24]. In particular, the homoharringtonine, a plant alkaloid derived from Cephalotaxus for-tune, and the halofuginone, a halogenated derivative of febrifugine, the active component of the medicinal herb Dichroa febrifuga, were identified as inhibitors of TMPRSS2 [24].

Furthermore, we found that zeaxanthin significantly inhibited the activity of both viral proteases, 3CL^pro^ and PL^pro^ (Figure 5). These data underscore zeaxanthin’s potential as a broad-spectrum protease inhibitor, a quality that is particularly valuable in the fight against rapidly mutating viruses like SARS-CoV-2. Indeed, once inside, the viral RNA is translated into a polyprotein, which undergoes autocleavage to release the 3C-like main protein protease (3CL^pro^). This protease cleaves the polyprotein to produce other proteins essential for viral replication. Thus, PL^pro^ and 3CL^pro^ play crucial roles in coronavirus infection and are prime targets for antiviral drug development.

Our data indicate that pistachio extracts and zeaxanthin possess significant antiviral properties against SARS-CoV-2. The different inhibition observed between α- and ο-variants suggests that these compounds may offer a targeted approach in combating specific SARS-CoV-2 new emergent variants. Mutations of the Omicron variant compared to the alpha variant play a key role in the zeaxanthin interaction, as demonstrated by MD simulation studies. However, further research is crucial to fully elucidate the involved molecular mechanisms and to evaluate the clinical relevance of these findings. This study paves the way for developing novel antiviral therapies based on natural compounds, which could complement existing treatments and enhance our ability to manage COVID-19 and future coronavirus outbreaks.

## 4. Materials and Methods

### 4.1. Cells

VERO cells and HEK-293T cells were cultured in Dulbecco’s Modified Eagle’s High Glucose Medium (DMEM, Euroclone, Pero, MI, Italy) supplemented with 6% fetal bovine serum (FBS) and 10% FBS, respectively. A549 cells (adenocarcinoma human alveolar basal epithelial cells) were kindly provided by Prof. Giorgio Gribaudo and cultured in HAM’S F-10 medium (Euroclone, MI, Italy) with 10% FBS and 2 Mm L-Glutamine. All mediums are supplemented with 100 U/Ml penicillin and 100 mg/Ml streptomycin. All cell lines were grown at 37 °C in a 5% CO_2_ incubator.

### 4.2. Materials

Californian natural raw polyphenols-rich extract (NRRE, Pistacchi Sgusciati California Pissgsu01/BSV1, L67316221262) and roasted unsalted polyphenols-rich extract (RURE, Pistacchi Sgusciati Tostati California Pissgstu01/BSV1) pistachio polyphenols-rich extracts were kindly supplied by the American Pistachio Growers (Fresno, CA, USA). NRRE or RURE (10 g) were extracted with *n*-hexane as previously described [19], and the quantification of phenolic compounds was performed by RP-HPLC-DAD.

### 4.3. Viability Assay

To assess cell viability following treatment with pistachio extracts, the ViaLight™ Plus Cell Proliferation and Cytotoxicity Bioassay (Lonza Group Ltd., Basel, Switzerland) was employed. A549 cells were seeded in 96-well plates and treated with varying concentrations of NRRE and RURE at 0.2, 0.3, 0.4, 0.6, 0.8, 1, and 2 mg/Ml, as well as zeaxanthin at 1, 5, 10, 20, and 25 µM for 72 h. After the treatment period, cell viability was measured using the GloMax^®^ Multi Microplate Luminometer (Promega Corporation, Madison, WI, USA) in combination with the ViaLight™ Plus assay kit, (Lonza, Basel, Switzerland)which quantifies light emission corresponding to ATP degradation.

The luminescence values were then converted Into a cell viability index (%) using Equation (1):Cell viability (%) = (*A* − *B*)/(*C* − *B*)(1)
where *A* is the average luminescence of the treated samples, *B* is the background luminescence, and *C* is the average luminescence of the untreated control samples.

### 4.4. Production and Characterization of Alpha and Omicron Pseudoviruses

A luciferase (Luc)-expressing pseudovirus containing the alpha and omicron S protein of SARS-CoV-2 was generated in the envelope-defective lentiviral backbone. The protocol aims to generate coronavirus (CoV-2) spike (S) fusion protein pseudotyped particles with a Lentivirus core and luciferase reporter using a simple transfection procedure of the widely available HEK-293T cell line. It is based on the transient, coordinated expression of an MLV expression construct and all necessary packaging proteins delivered into producer cells by simultaneous transfection with lentiviral expression and packaging plasmids [25,26]. One plasmid encodes the MLV core genes gag and pol. The second plasmid is a transfer vector that encodes a firefly luciferase reporter gene, an MLV MJ-RNA packaging signal, and 5′- and 3′-flanking MLV long terminal repeat (LTR) regions. The third plasmid, pcDNA3.1(−) SARS-Swt-C9, kindly provided by Jean K. Millet, encodes the Spike glycoprotein of the alpha variant. Similarly, the recombinant plasmid bearing the S protein of the SARS-CoV-2 Omicron variant is cloned on the PunO1 vector and contains the Spike coding sequence from the Omicron SARS-CoV-2 variant (BA.2 lineage), (Catalog code: p1-spike-v12). The PunO1-SpikeV12 was transfected into HEK-293T cells to generate the ο-SARS-CoV-2 pseudotyped virus. The co-expression of these three plasmids leads to the synthesis of MLV capsid proteins, spike envelope protein, and LTR-flanked luciferase, which assemble on the plasma membrane and generate pseudotyped particles. The supernatants containing SARS CoV-2 pseudovirus were harvested 72 h after transfection, and the pseudovirions enrichment was performed via PEG-it™Virus Precipitation which concentrates the pseudotype particles (PEG Virus Precipitation Kitab102538) (Cambridge Biomedical Campus, Cambridge, UK). The production efficiency was measured by measuring the luciferase gene expression using the Luciferase Assay System (Promega, Madison, WI, USA) according to the manufacturer’s instructions [26].

The luciferase activity was measured in Vero cells 72 h post-infection with no envelope pseudotyped particles (Δenvpp), VSV-G-pseudotyped particles (VSV-Gpp), and α and ο-SARS-CoV-2 S-pseudotyped particles (SARS-Spp). The procedure was previously reported [26].

### 4.5. In Vitro Pseudovirus Binding Assay

For the pseudovirus entry inhibition screening, α and ο-SARS-CoV-2 spike pseudotyped virus particles were pretreated with 0.6 mg/ML of either NRRE or RURE and 10 ΜM of zeaxanthin for 1 h at 4 °C. The pretreated pseudovirus was then added to A549 cells seeded in a 96-well plate. The transduction was performed at 4 °C for 2 h, followed by washing and replacing with a growth medium. The luciferase assay was carried out 72 h post-transduction. The cell lysis buffer reagent (Promega, Madison, WI, USA) was added to each well after removing the media, and plates were incubated with shaking for 10 min; 100 ML of luciferin substrate was added to each well, and luminescence was read with 1 min integration and delay time. The assay was performed in triplicate, and the data were reported as RLU compared to the uninfected control.

### 4.6. In Vitro Pseudovirus Entry Inhibition Assay

A549 cells (4 × 104) were seeded in a 96-well plate in 100 ML of HAM’S F-10 medium containing 10% FBS and treated with 10 ΜM of zeaxanthin for 2 h at 37 °C. After treatment, the cells were transduced separately with α- and o-SARS-CoV-2 spike pseudotyped virus particles. After 72 h, the cells were lysed and subjected to luciferase activity assay and western blot analysis.

### 4.7. Western Blot Analysis

Immunoblot analysis was carried out to evaluate the accumulation of TMPRSS2, as previously reported [27]. Briefly, a total of A549 cell lysates were prepared from cells using SDS sample buffer 1x (62.5 MM Tris-HCl (Tris(hydroxymethyl) aminomethane hydrochloride) PH 6.8; 50 MM DTT (dithiothreitol); 10% glycerol; 2% SDS (sodium dodecyl sulfate); 0.01% Bromophenol Blue; EDTA-free Protease Inhibitor Cocktail 1x (Roche, Basel, Switzerland) and were then boiled for 5 min. An equal amount of protein extract was loaded onto a 10% sodium dodecyl sulfate-polyacrylamide gel, transferred to nitrocellulose membranes, and blocked at 4 °C overnight in 5% non-fat dry milk-TBS. GAPDH (sc-32233, Santa Cruz, CA, USA) and TMPRRS (sc-515727, Santa Cruz, CA, USA) were detected by secondary HRP conjugated goat anti-mouse IgG (Merk Millipore, Burlington, VT, USA). Specific bands were visualized using LiteUP WB Chemiluminescent Substrate (Euroclone, MI, Italy). Quantitative densitometry analysis of immunoblot band intensities was performed using ImageJ software (ImageJ Version 1.54). The intensity of the target protein was divided by the intensity of the GAPDH and graphically represented by GraphPad Prism 6 software (GraphPad Software, San Diego, CA, USA).

### 4.8. SARS-CoV-2 3CL^pro^ and PL^pro^ Luminescent Assays

The inhibitory activity of zeaxanthin against 3CLpro and PLpro was determined using the SARS-CoV-2 3CLpro and PLpro Luminescent Assays, respectively, as reported by the manufacturer instructions (CS331201-SARS-CoV-2 3CLpro and PLpro Luminescent Assays, Promega, Madison, WI, USA) [20,28]. Briefly, purified recombinant SARS-CoV-2 3CLpro (9000 µg/mL) and GST-PLpro (1000 nM) were tested against zeaxanthin (10 µM), with GC376 as a positive control at 2 µM and CAMOSTAT (5 µM, inhibitor of the serine protease TMPRSS2). The assays were performed in 96-well plates. For 3CLpro, 12.5 µL of the enzyme (16 µg/mL) was incubated with 12.5 µL of assay buffer and zeaxanthin at 37 °C for 60 min. The substrate was then added to a final concentration of 40 µM. Luminescence was measured after 20 min using the Luciferin Detection Reagent. Similarly, PLpro (10 nM) was incubated with zeaxanthin at 25 °C for 30 min in an assay buffer, followed by substrate addition. Luminescence was measured after 10 min using the Luciferin Detection Reagent.

### 4.9. Molecular Docking

Flexible ligand docking experiments were performed by employing AutoDock 4.2.6 software implemented in YASARA (v. 24.10.24, YASARA Biosciences GmbH, Vienna, Austria), using the crystal structure of the SARS-CoV-2-RBD alpha variant complexed with ACE2 has been extracted as a monomer from the Protein Data Bank PDB code 6M0J and PDB code 7WPB for SARS-CoV-2 Omicron Variant RBD complexed with ACE2 (PDB, http://www.rcsb.org/pdb) (accessed on 27 October 2024). The structure of the Alpha variant was generated by adding the three mutations located in the RBD region to the WT-RBD PDB using YASARA. In particular, the procedure followed to generate mutated proteins was (i) replace the side chain atoms of the residue to be mutated from the WT-RBD PDB (N501Y, Y453F, and N439K); (ii) run the YASARA cleans tool to automatically add the missing side chain atoms of the new residue. The maps were generated by the program AutoGrid (4.2.6) with a spacing of 0.375 Å and dimensions encompassing all atoms extending 5 Å from the surface of the structure of the crystallized ACE2. Point charges were initially assigned according to the AMBER03 force field and then damped to mimic the less polar Gasteiger charges used to optimize the AutoDock scoring function. The structure of all ligands was optimized at the semiempirical level of PM6 theory [29]. As previously reported, all parameters were inserted at their default settings [30]. In the docking tab, the macromolecule and ligand were selected, and GA parameters were set as ga_runs = 100, ga_pop_size = 150, ga_num_evals = 25,000,000, ga_num_generations = 27,000, ga_elitism = 1, ga_mutation_rate = 0.02, ga_crossover_rate = 0.8, ga_crossover_mode = two points, ga_cauchy_alpha = 0.0, ga_cauchy_beta = 1.0, number of generations for picking worst individual = 10.

### 4.10. Molecular Dynamics Simulations

The molecular dynamics simulation studies were carried out according to the protocols described in our previous studies [31,32]. The protein-ligand complex was placed in a simple point charge water-modeled periodic cubic box (8 Å) [33]. NaCl ions (0.9%) were added to mimic physiological conditions. An excess salt was added to neutralize the charge, while water molecules were deleted to readjust the solvent density to 0.997 g/mL. The AMBER force field was used, and a pressure of 1 atm was applied. Simulations were run at 298 K [34,35]. The final dimensions of the box were approximately 90 × 90 × 90 Å3. A short MD simulation was performed on water only to remove interferences. The box was brought to an energy minimum using the steepest descent minimization to remove conformational stress until convergence (<0.01 kcal/mol Å). Each simulation lasted 200 ns, and the ligand-protein contacts during the MD were recorded every 200 ps. The entire simulation protocol, including the initial energy minimization steps, is managed by ‘macros’ in YASARA Structure. A graphical user interface facilitates running the simulation through these macros. In this study, the macro ‘md_run.mcr’ was used to ensure seamless execution of the simulation commands, while the trajectory analysis and binding energy during the simulation were performed using ‘md_analyze.mcr’. In addition to global dynamic analysis, the binding free energies of the selected protein-ligand complexes were determined using the Molecular Mechanics Poisson-Boltzmann Surface Area (MM-PBSA) method. The analysis was performed using the ‘md_analyzebindenergy.mcr’ macro from YASARA Structure. These macros contained YASARA commands, and the YANACONDA macro language was employed for more complex tasks. Ligand–protein contacts occurring during MD were analyzed using the function of YASARA software (v. 23.5.19, YASARA Biosciences GmbH, Vienna, Austria).

### 4.11. Statistical Analysis

Results are shown as the mean/SD based on a minimum of three independent experiments. Statistical analysis utilized GraphPad Prism 8.0.1.244 software (GraphPad Software Inc., San Diego, CA, USA) and involved one-way analysis of variance (ANOVA). Significance levels of *p*-values are represented by asterisks (**, ***, ****), indicating values less than 0.01, 0.001, and 0.0001, respectively. The EC_50_ values of inhibitors were calculated from dose inhibition curves using nonlinear regression analysis by GraphPad Prism.

## 5. Conclusions

Our results indicated that pistachio extracts and zeaxanthin possessed a selective inhibitory effect on the entry of specific SARS-CoV-2 variants, with a notable reduction in the ability of the omicron variant to enter host cells. Interestingly, zeaxanthin did not exhibit the same inhibitory effect on the alpha variant’s entry. However, zeaxanthin appears to modulate host cell factors, mainly by reducing the expression of TMPRSS2, a protease involved in the entry of the alpha variant. This suggests that while zeaxanthin may not directly hinder the alpha variant’s binding, it still offers protective effects by reducing the availability of this crucial host factor for viral entry.

Molecular modeling and MD simulations revealed zeaxanthin’s stronger interaction with the o-RBD than the alpha variant, showing greater stability and favorable binding affinity (−8.77 kcal/mol). This correlates with reduced viral entry in vitro, suggesting zeaxanthin destabilizes RBD-ACE2 binding.

Furthermore, the observation that zeaxanthin is a potent inhibitor of viral proteases strengthens its potential as a broad-spectrum antiviral agent. By targeting both viral entry mechanisms and protease activity, zeaxanthin provides a multifaceted approach to preventing viral infection. These findings highlight its potential for emerging specific variants and as a general therapeutic agent against SARS-CoV-2, capable of offering protection across different variants. The dual inhibitory action of zeaxanthin on both the virus and host factors associated with viral entry underscores its promise as an effective treatment strategy for managing COVID-19, particularly in the face of evolving variants.

Lastly, zeaxanthin’s ability to inhibit viral entry and reduce the expression of critical host factors, combined with its effectiveness against viral proteases, positions it as a valuable broad-spectrum inhibitor. Future studies should explore its potential in clinical settings, particularly in combination therapies, to fully assess its efficacy in managing infections caused by various SARS-CoV-2 variants.

## Figures and Tables

**Figure 1 ijms-26-01667-f001:**
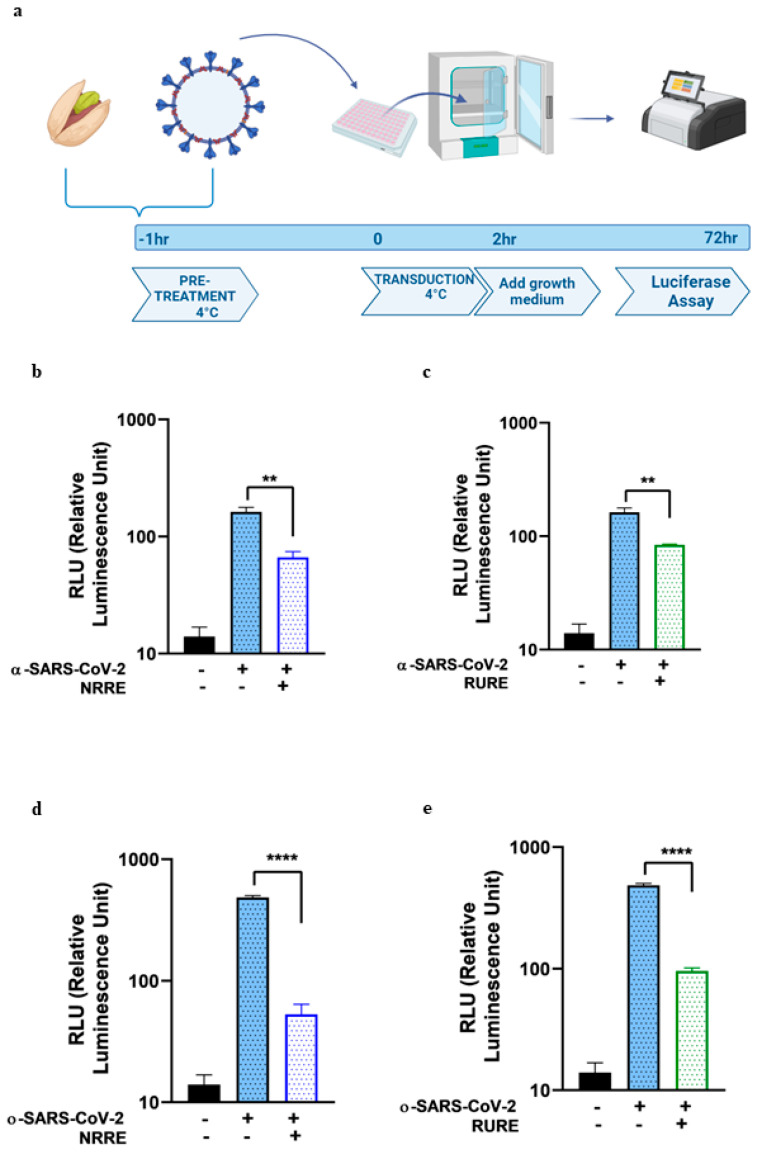
In vitro pseudovirus binding assay of NRRE and RURE using α-SARS-CoV-2 and ο-SARS-CoV-2 pseudotyped particles on A549 cells. (**a**) Schematic diagram of the binding assay. (**b**,**c**) The luciferase activity was performed in A549 cells 72 h post-infection with or without NRRE (0.6 mg/mL) or RURE (0.6 mg/mL) using α-SARS-CoV-2. (**d**,**e**) The luciferase activity was performed in A549 cells 72 h post-infection with or without NRRE (0.6 mg/mL) or RURE (0.6 mg/mL) using ο-SARS-CoV-2 pseudotyped particles. Experiments were performed in triplicates, and the data presented the RLU of three independent experiments (n = 3). Error bars indicate standard deviation (SD). The asterisks (** and ****) indicate the significance of *p*-values less than 0.01 and 0.0001, respectively, compared to untreated infected control.

**Figure 2 ijms-26-01667-f002:**
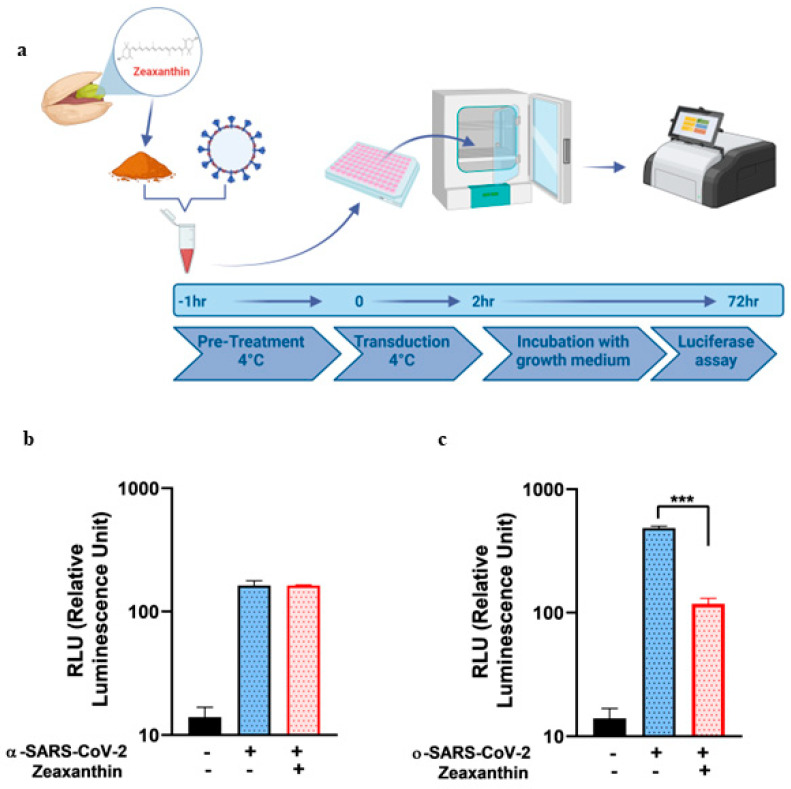
In vitro pseudovirus binding assay using zeaxanthin against α-SARS-CoV-2 and ο-SARS-CoV-2 pseudotypes. (**a**) Schematic diagram of the binding assay. (**b**,**c**) The luciferase activity was performed in A549 cells 72 h post-infection with or without zeaxanthin (10 µM) using α and ο-SARS-CoV-2 S-pseudotyped particles. Experiments were performed in triplicates, and the data presented the RLU of three independent experiments (n = 3). Error bars indicate standard deviation (SD). The asterisks (***) indicate the significance of *p*-values less than 0.001 compared to untreated infected α and ο-SARS-CoV-2 pseudotyped particles.

**Figure 3 ijms-26-01667-f003:**
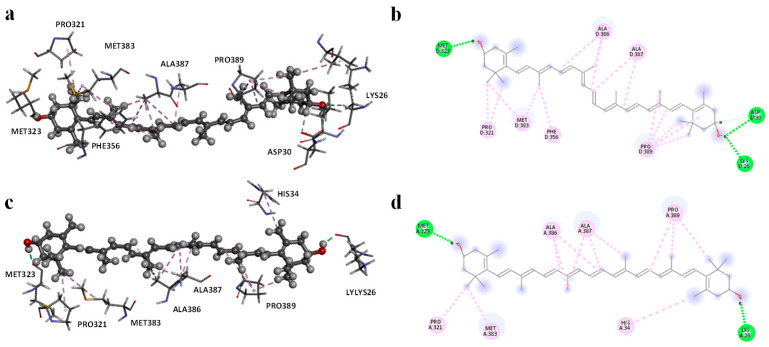
Interaction profile of the docked poses (**a**) and 2D diagram interaction profile (**b**) of zeaxanthin/α-RBD complex and zeaxanthin/ο-RBD complex (**c**,**d**).

**Figure 4 ijms-26-01667-f004:**
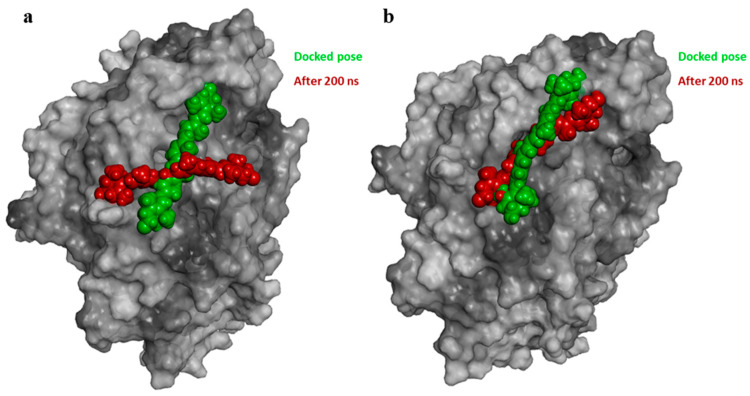
View of the hydrophobic surface (grey) of the α-RBD (**a**) and ο-RBD (**b**) domain and docked zeaxanthin before (green) and after 200 ns of MD simulation (red).

**Figure 5 ijms-26-01667-f005:**
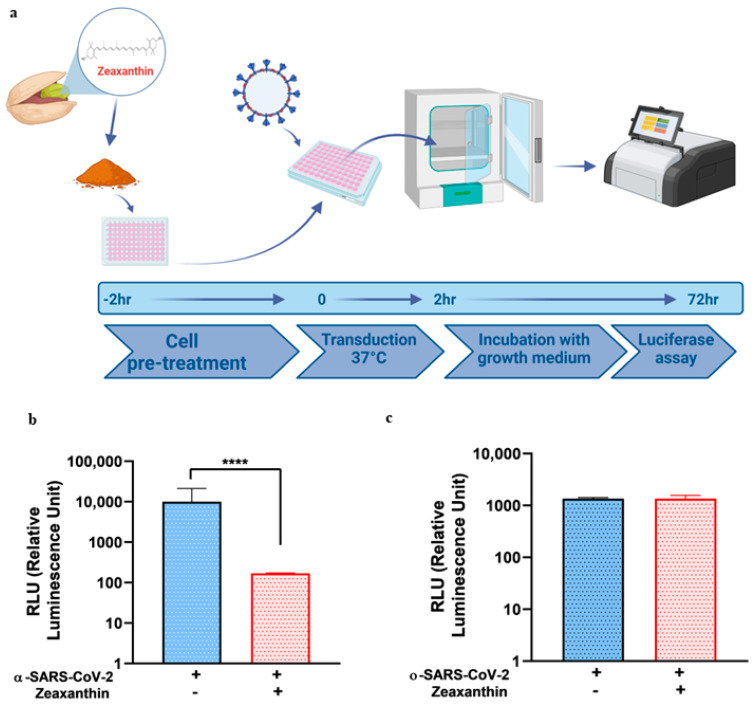
In vitro pseudovirus entry inhibition assay following zeaxanthin treatment. (**a**) Schematic diagram of pseudovirus entry inhibition assay. (**b**,**c**) The luciferase activity was performed in A549 cells 72 h post-infection with or without zeaxanthin (10 µM) using α-SARS-CoV-2 and ο-SARS-CoV-2 pseudoviruses. Experiments were performed in triplicates, and data represented the RLU of three independent experiments (n = 3). Error bars indicate standard deviation (SD), and **** indicates the significance of *p*-values less than 0.0001.

**Figure 6 ijms-26-01667-f006:**
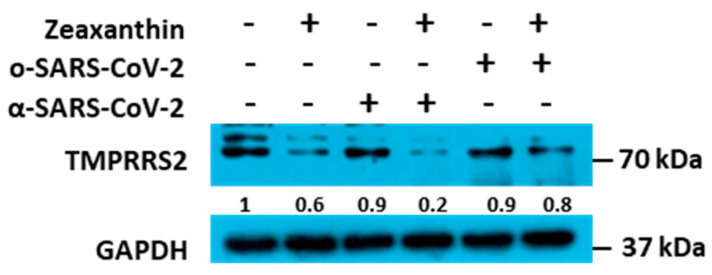
Expression of TMPRSS2 protein in zeaxanthin-treated and SARS-CoV-2-transducted cells. After zeaxanthin treatment (10 µM), A549 cells were transduced or not with α-SARS-CoV-2 and ο-SARS-CoV-2 pseudoviruses, and accumulation of TMPRSS2 protein was evaluated. GAPDH was used as a housekeeping gene. Quantification of the TMPRSS2 bands was calculated by the ImageJ Version 1.54.

**Figure 7 ijms-26-01667-f007:**
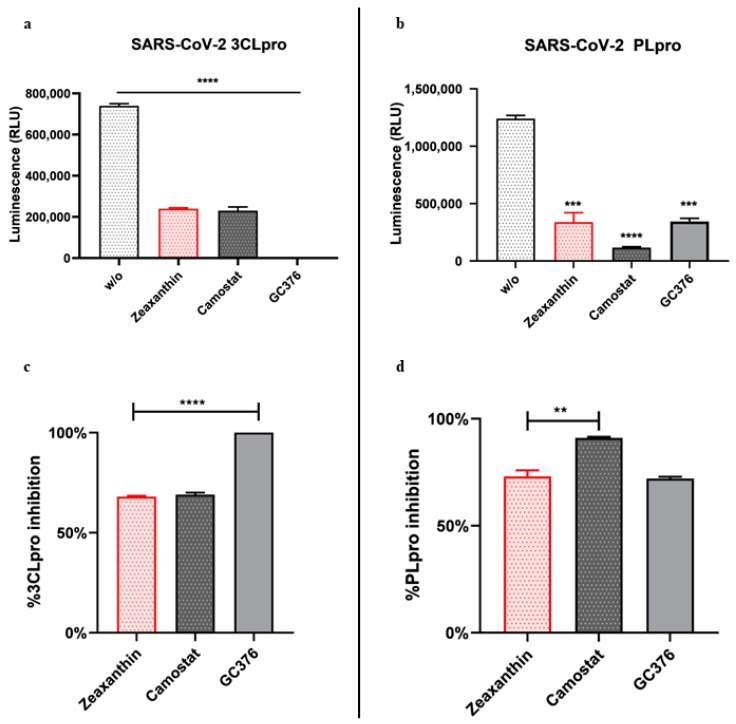
In vitro inhibition of SARS-CoV-2 3CL^pro^ and PL^pro^. Purified recombinant SARS-CoV-2 3CL^pro^ and SARS-CoV-2 PL^pro^ were combined with zeaxanthin (10µM) and added to 3CL^pro^ substrate solution for 60 min at 37 °C. GC376 and Camostat were used as positive protease inhibitor controls. The reaction was stopped by adding Luciferin Detection Reagent, and the luminescence was recorded on a plate-reading luminometer. Results were expressed as percentages of RLU (**a**,**b**) and 3CL^pro^ and PL^pro^ inhibitory activity (**c**,**d**). ** *p* < 0.01, *** *p* < 0.001, **** *p* < 0.0001.

## Data Availability

The data presented in this study are available on request from the corresponding author.

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
