# Peer review of "Selective Control by Pistacia vera L. and Its Carotenoid Zeaxanthin on SARS-CoV-2 Virus"

_ijms, 2025, doi:10.3390/ijms26041667_

Round 1
Reviewer 1 Report (Previous Reviewer 2)
Comments and Suggestions for Authors
This manuscript is significantly improved from its last version.
Author Response
We would like to thank you for taking the time to review our manuscript.
Reviewer 2 Report (New Reviewer)
Comments and Suggestions for Authors
The authors present the activity of a compound extracted from pistachios that could significantly inhibit SARS-CoV-2 entry by blocking the binding of the alpha and omicron variants. Moreover, its effect appears to be more potent on the RBD domain of the omicron spike (S) protein, reducing viral entry, while its interaction with the alpha variant was weaker. The combination of experimental results and molecular dynamics simulations allows the proposal of action mechanisms for the compound and provides a molecular-level description of the phenomenon. Based on this, the article is suitable for publication with minor revisions, which I detail below:
1. The term NRRE, RUE, RLU must be explained the first time they are mentioned in the text. Revise the complete text
2. Line 197. In Figure 4 it is indicated that the green structure is at simulation time 0. Does this structure correspond to the one obtained by docking or is it the one that results prior to the production stage? It is suggested to indicate it in the legend of the figure and to modify “after 0” by start simulation .
3. Line 522. In methodologies point 4.10 specify if carbohydrates were considered and if so how the parameterization was performed. Also, describe in more detail the preparation of the system, which should include structural optimizations, heating and balancing before the production stage.
4. Since they already have the simulation, it would be interesting to calculate deltag with MMGBSA to compare between the analyzed variants. These results could be included in a table with the Docking energy in supplementary material and include a comment in discussions on the results obtained.
Author Response
We sincerely thank the Reviewers for their insightful comments, which have greatly contributed to improving the quality of our manuscript. We have systematically revised the manuscript to address all concerns in response to your feedback. Below, we provide a point-by-point response to your comments, with the original comments in boldface and our responses in regular typeface. We are resubmitting a revised version of the manuscript with several corrections and enhancements compared to the previous version.
- The term NRRE, RUE, RLU must be explained the first time they are mentioned in the text. Revise the complete text.
We thank the Reviewer for their valuable feedback. We confirm that the terms NRRE (Natural Raw Polyphenols-Rich Extract) and RURE (Roasted Unsalted Polyphenols-Rich Extract) were defined upon their first mention in the text (lines 107 and 108). The term relative luminescence unit (RLU) has been removed from the text except for its first appearance in line 113.
- Line 197. In Figure 4 it is indicated that the green structure is at simulation time 0. Does this structure correspond to the one obtained by docking or is it the one that results prior to the production stage? It is suggested to indicate it in the legend of the figure and to modify “after 0” by start simulation.
We thank the Reviewer for the suggestion. We edited the manuscript accordingly.
- Line 522. In methodologies point 4.10 specify if carbohydrates were considered and if so how the parameterization was performed. Also, describe in more detail the preparation of the system, which should include structural optimizations, heating and balancing before the production stage.
We thank the Reviewer for the suggestion. We edited the manuscript accordingly.
- Since they already have the simulation, it would be interesting to calculate deltag with MMGBSA to compare between the analyzed variants. These results could be included in a table with the Docking energy in supplementary material and include a comment in discussions on the results obtained.
We sincerely appreciate the Reviewer’s valuable suggestion. In response, we have revised the manuscript accordingly. The free energies of binding of the selected protein-ligand complexes were computed using the Molecular Mechanics Poisson-Boltzmann Surface Area (MM-PBSA) method. This analysis was performed using the md_analyzebindenergy.mcr macro in YASARA Structure. To provide a comprehensive comparison, we generated graphs (Figure S9) illustrating the free energy of binding trends obtained via the MM-PBSA method alongside the free energy of binding profiles derived from the docking method throughout the MD simulation.
This manuscript is a resubmission of an earlier submission. The following is a list of the peer review reports and author responses from that submission.
Round 1
Reviewer 1 Report
Comments and Suggestions for Authors
This study examined the antiviral effects of pistachio extracts (NRRE and RURE) and zeaxanthin against SARS-CoV-2, as well as the differential inhibition observed between the α- and ο-variants. I have two major concerns regarding the experimental design and the reliability of the results, as outlined below:
- Mature TMPRSS2 proteolytically processes the SARS-CoV-2 spike protein bound to the ACE2 receptor, facilitating membrane fusion and viral entry. In the binding inhibition assays, why did you pretreat the pseudotype variants with the compounds first instead of pretreating the A549 cells, given that the inhibition mechanism for these compounds may involve the cell membrane enzyme TMPRSS2?
- It appears that a molecular weight ladder was not used in the western blot, as shown on the nitrocellulose membrane (in the supplementary document). How can you identify the TMPRSS2 protein bands when there are many bands near the expected molecular weight showing nonspecific antibody bindings? Additionally, the two pieces of nitrocellulose membrane do not seem to come from an intact original membrane, as the cutting edges do not match on both ends.
Further questions about the manuscript writing:
- Page 1, Line 34: You introduced alpha and beta coronaviruses and then transitioned to SARS-CoV-2. To improve the logical flow, consider adding that SARS-CoV-2 belongs to the betacoronavirus group.
- Page 1, Lines 45-47: No reference is provided to support the statement.
- Page 2, Line 58: The phrase “nearly impossible” is too absolute.
- Page 2, Line 62: A punctuation mark is missing.
- Page 2, Line 89: At the first mention of “NRRE and RURE,” the manuscript suggests checking the Materials and Methods section for more information. However, no explanation of these abbreviations or details on their preparation is provided in that section.
- Page 3, Line 100: To be accurate, add "on A549 cells" to the end of "In vitro pseudovirus binding assay of NRRE and RURE using α-SARS-CoV-2 and ο-SARS-CoV-2 pseudotyped particles."
- Page 4, Line 112: There is a grammatical error in “we investigated whether could exhibit a similar inhibitory effect on SARS-CoV-2 by …” The sentence should read, "whether it could exhibit a similar inhibitory effect."
- Page 5, Line 133: Please be more specific about what you aim to explain in “To further explain our data…”
- Page 5, Lines 139-140: “The results indicate that zeaxanthin specifically inhibits alpha entry but not the Omicron variant (Figure 3b vs. 3c), suggesting an effect on TMPRSS2-mediated internalization.” Why does this difference in inhibition effects on alpha and Omicron variants relate to TMPRSS2-mediated internalization? Please provide more explanation.
- Page 6, Line 159: Change “First, we treated or not the A549 cells for two hours with zeaxanthin…” to “First, we treated the A549 cells with or without zeaxanthin for two hours…”
- Page 7, Line 201: In “Similarly, zeaxanthin significantly reduced papain-like protease PLpro, as Figures 5b and 5d reported,” do you mean "reduced the activity of papain-like protease PLpro"?
- Page 8, Lines 224-228: The section discussing the roles of TMPRSS2 in viral entry to the host cell should be included in the introduction to facilitate a better understanding of the results.
Author Response
We sincerely thank the reviewer for the comments which were of great help in revising the manuscript. Accordingly, the revised manuscript has been systematically improved. In attached, you will find a point-by-point description which includes the original reviewer comments in boldface and the responses in red regular typeface. We specify that we resubmit the new version of the manuscript with several corrections compared to the previous version.
Reviewer 2 Report
Comments and Suggestions for Authors
As described in this manuscript, pistachio-derived zeaxanthin inhibited the expression of host protease TMPRSS2 at protein levels, limiting the internalization of α-SARS-CoV-2 but not the ο-SARS-19 CoV-2 variant, highlighting the different entry mechanism of the two variants. Some major problems listed below should be addressed.
1. I cannot see the full names of NRRE and RUREF when they first appear in the manuscript.
2. As shown in Figure 1, NRRE and RUREF both could inhibit entry of both ο-SARS-CoV-2 and α-variant, but why in Figure 2 and 3, when using the antiviral ingredient zeaxanthin from NRRE and RUREF, only one variant was inhibited? Are there other unidentified ingredients in the extracts inhibiting the both viral strains entrance?
3. In abstract, “limiting the internalization of α-SARS-CoV-2 but not the ο-SARS-CoV-2 variant”, but in Figure 2 “The data obtained revealed that zeaxanthin significantly inhibits the binding of the ο-SARS-CoV-2 but not the α-variant (Figure 2, panel b vs c)”, and in Figure 3 “The results showed that zeaxanthin specifically inhibits the alpha entry but not the omicron variant (Figure 3b vs 3c)”, these statements seem to be contradictory. Please clarify in the text.
4. Also in line 140 of page 5, “suggesting that it could affect the TMPRSS2-mediated internalization”, as we all know, ο-SARS-CoV-2 entry is dependent on Cathepsin L, whereas the α-variant is dependent on Furin and TMPRSS2. They should check whether the activities or expression levels of these three human proteases are affected by zeaxanthin, but not just checking the reduced level of TMPRSS2 by the treatment of zeaxanthin (Figure 4).
5. They showed that zeaxanthin inhibited 3CLpro and PLpro. However, these two proteases are not involved in virus entry into A549. They should check the activities and expression levels of Cathepsin L, Furin, and TMPRSS2, but not 3CLpro and PLpro.
6. Could zeaxanthin also block RBD:ACE2 interaction, which is essential for attachment of SARS-CoV-2 to host? Since ο-SARS-CoV-2 and α-variant have different RBD sequences, so could inhibit RBD:ACE2 interaction differently, this might be the reason that zeaxanthin inhibit ο-SARS-CoV-2 and α-variant viral entry differently. Please check.
7. They claim in Discussion that “These findings are consistent with our previous data obtained on HSV-1 entry [19] but also reinforce the reliability of our research”. However, HSV-1 and SARS-CoV-2 use totally different ways (different proteins involved) for entry. The compound which can inhibit HSV-1 entry is not necessary to inhibit another virus entry. The statement could be revised to “The natural product zeaxanthin that we showed previously to inhibit HSV-1 entry [19] also works as SARS-CoV-2 entry inhibitor, although the targets and mechanisma are different”.
Author Response
We sincerely thank the reviewer for comments which were of great help in revising the manuscript. Accordingly, the revised manuscript has been systematically improved. In attached, you will find a point-by-point description which includes the original reviewer comments in boldface and the responses in red regular typeface. We specify that we resubmit the new version of the manuscript with several corrections compared to the previous version.
Round 2
Reviewer 1 Report
Comments and Suggestions for Authors
The corrections and responses have addressed all my questions in the reviewer report. The manuscript is now in good shape for publication.
Reviewer 2 Report
Comments and Suggestions for Authors
The authors did not do any experiment to answer my concerns, which are critical issues to be addressed.